# Integrative Management of Metabolic Syndrome in Youth Prescribed Second-Generation Antipsychotics

**DOI:** 10.3390/medsci8030034

**Published:** 2020-08-17

**Authors:** Jessie Rice, Ujjwal Ramtekkar

**Affiliations:** 1Department of Psychiatry, University of Arizona, Tucson, AZ 85721, USA; ricej@email.arizona.edu; 2Partners for Kids, Columbus, OH 43215, USA; 3Department of Psychiatry and Behavioral Health, The Ohio State University College of Medicine, Columbus, OH 43210, USA; 4Nationwide Children’s Hospital, The Ohio State University College of Medicine, Columbus, OH 43210, USA

**Keywords:** antipsychotic, obesity, metabolic syndrome, diabetes, pediatric

## Abstract

Weight gain and metabolic syndrome are common side effects of second-generation antipsychotics and carry significant health consequences both in childhood and into adulthood. This review highlights evidence-based, non-pharmacologic interventions to assist in the management of these side effects. Such intervention categories include dietary, physical activity, sleep, stress management, and nutritional supplementation. Interventions with the highest quality evidence include increasing the consumption of fruits, vegetables, and whole grains, increasing physical activity, improving sleep, and fish oil supplementation. We suggest that clinicians work with patients on managing metabolic side effects in a patient-centered way, incorporating principles of motivational interviewing, to reduce the risk of metabolic syndrome.

## 1. Introduction

Pediatric obesity is a major public health problem. According to the Centers for Disease Control (CDC), 20.6% of adolescents (ages 12–19) are obese. Amongst school-aged children (ages 6–11) and preschoolers (ages 2–5), obesity rates are at 18.4% and 13.9%, respectively [1]. Pediatric obesity is associated with numerous medical complications including type 2 diabetes, hypertension, dyslipidemia, nonalcoholic fatty liver disease, as well as the risk of cancer in adulthood [2]. The term “metabolic syndrome (MetS)” arose to describe the clinical phenomenon of some of these complications, specifically central obesity, insulin resistance, hypertension, and dyslipidemia [2]. Specific cut-off values defining each of these features in youth, however, have not been definitively established [2]. Pediatric MetS is also strongly predictive of adult MetS and type 2 diabetes [3]. Additionally, non-alcoholic fatty liver disease (NAFLD) is associated with MetS in adults and children, and it is unclear if NAFLD is a precursor to or a manifestation of MetS [4,5].

Pediatric obesity is also linked to mental health problems, even after correcting for socioeconomic status differences. For example, obese children are more likely to have externalizing problems, grade repetition, and school problems. An association exists between obesity and ADHD, depression, and learning disabilities [6].

Pediatric obesity and MetS are major issues in the prescribing of certain psychotropic medications, particularly second-generation antipsychotics (SGA) due to significant weight gain. Youth are at higher risk of antipsychotic-induced weight gain than their adult counterparts [7]. This presents a challenging situation for prescribing clinicians. Given the complexities of pediatric obesity in general, with its numerous genetic, environmental, and medication-related factors, there are multiple intervention points. The same framework can be extended in the setting of SGA-induced obesity. While there are several studies and reviews encompassing adjunctive pharmacologic management of MetS in youth, to our knowledge, there have been no reviews combining the specific pathophysiology of SGA-related MetS and patient-centered lifestyle interventions [8,9,10,11]. Non-pharmacologic interventions for MetS in youth is the focus of our paper. Consensus guidelines from the American Medical Association (AMA) and CDC, endorsed by the American Academy of Pediatrics (AAP), recommend starting with non-pharmacologic lifestyle interventions in the management of pediatric obesity [12]. In this review, we deliver to clinicians prescribing SGAs to youth practical, evidence-based suggestions on lifestyle issues including diet, physical activity, sleep, stress management, and targeted nutritional supplementation in certain cases, all from a patient-centered approach. A review of pharmacologic and surgical options for the management of pediatric MetS is beyond our scope.

## 2. Epidemiology of Antipsychotic-Related Metabolic Syndrome

Currently, Federal Drug Administration-approved indications for antipsychotic prescription in youth are autism-related irritability, bipolar I disorder, and schizophrenia. Despite this, most antipsychotic prescription in youth is off-label. In the United States, it is estimated that around 66–80% of SGAs are prescribed off-label to youth with common diagnoses being Attention-Deficit/Hyperactivity Disorder (ADHD) and depression [13,14]. Among developed countries, the United States has the highest rates of antipsychotic prescription in youth [15]. Since the 1990s, rates of antipsychotic prescription in youth have somewhere between doubled and quadrupled [16]. SGA prescription in youth carries a higher risk of side effects than in the adult population in general, including weight gain [7]. Youth prescribed SGAs are also much less likely than their adult counterparts to undergo recommended monitoring of metabolic parameters [14].

Of the various longitudinal studies involving youth taking SGAs, weight gain occurs commonly. In a European trial, 53.4% of youth prescribed an SGA experienced a metabolic side effect, most commonly within the first three months of starting the medication [14]. The average weight gain over a twelve-month period was around 5 kg, with body mass index (BMI) increasing by 1.5 points [14]. Clinically significant elevations in cholesterol occurred in about 36%, and diabetes developed in about 7% in this study. A longitudinal study done in the United States between 2001 and 2007, the SATIETY trial, found that after only 10.8 weeks of SGA exposure, weight increased by 4.4–8.5 kg, depending on the medication used [17]. The highest weight gain was associated with olanzapine (8.5 kg), followed by quetiapine (6.1 kg), risperidone (5.3 kg), and finally aripiprazole (4.4 kg). Serum total cholesterol increased by 15.6 mg/dL (both low-density lipoprotein (LDL), and high density lipoprotein (HDL)) and triglycerides increased by 24.3 mg/dL. Another cohort study between 2004 and 2006 of children ages 6–17 in a three-state Medicaid registry found that lipid disorders and diabetes were twice as common in SGA-treated youth than in the control group. Unfortunately, only 31.6% of this sample received glucose screening and only 13.4% received lipid screening [18].

Obesity is also correlated with higher risk of depressive symptoms [19,20,21]. In addition to psycho-social factors, there is underlying inflammation, and specifically neuro-inflammation. Central adiposity turns on a cascade of inflammatory signals like tumor necrosis factor-alpha (TNF-α), interleukin-1 beta (IL-1β), IL-6, and C-reactive protein that can negatively affect mood and lead to depressive symptoms [19,20,21].

## 3. Mechanisms of Metabolic Syndrome

The mechanisms underlying SGAs and their influence on the development of MetS are many and complex. In general, the higher the affinity to 5-hydroxytryptamine receptor 2C (5-HT2C) and histamine H1 (H1) receptors, the more significant the metabolic effects, as in the case of olanzapine and clozapine [22]. Much of the literature presented in this section comes from studies of healthy volunteers and animal studies, as it can be difficult to differentiate between metabolic changes brought about by a given psychiatric disorder versus a psychiatric medication (i.e., depression causing a youth to become more physically inactive, eat a less healthy diet, and therefore gain weight).

Antipsychotics are known to increase appetite, which leads to weight gain [22]. There are likely multiple mechanisms for this. One mechanism is via the antipsychotic’s affinity to central histamine H1 receptors, which exert subtle but important effects on appetite regulation and brown fat metabolism [23]. Some antipsychotics have more H1 affinity than others and therefore higher risk of weight gain. Another mechanism is through enhancing cell signaling when the hunger hormone, ghrelin, binds to its receptor [24].

SGAs can also disturb lipid profiles. Olanzapine and clozapine are known to increase triglycerides the most [25]. They increase de novo triglyceride synthesis in animal models and also up-regulate transcription factors sterol regulatory element-binding proteins (SREBP)-1 and 2, which regulate triglyceride and cholesterol synthesis [25]. Triglycerides increase in the presence of insulin-resistance due to hypersecretion of insulin, which again activates SREBP-1 and renders the body unable to inhibit the synthesis of the very low-density lipoprotein (VLDL) particles [25].

SGAs negatively impact glucose tolerance, which leads to the development of insulin resistance and type 2 diabetes mellitus [22,26]. Evidence from animal models as well as human studies suggests that the mechanisms by which SGAs contribute to insulin resistance are complex. It may occur on the level of the liver, the pancreatic beta-cells or even the autonomic nervous system [26]. Even a single dose of olanzapine in healthy adult volunteers was shown to raise fasting glucose over several hours compared to placebo [27]. Exercise may help protect against this effect: A mouse study found that “exhaustive but not moderate” exercise attenuated olanzapine’s effect on raising blood glucose [28]. This effect was mediated through interleukin-6 (IL-6), suggesting that raising IL-6 in this way may be protective against hyperglycemia. Another study found that olanzapine-treated rats who exercised regularly developed less glucose intolerance that their more sedentary control counterparts [29]. Glucose Transporter 4 (GLUT4) levels were expressed in higher levels in the skeletal muscle of the more active rats; it is known that GLUT4 is an insulin-dependent transporter of glucose into cells [29]. Human studies are needed to verify if exercise can attenuate SGA-induced hyperglycemia.

There are emerging data suggesting that baseline obesity may further complicate SGA-induced glycemic effects. A study utilizing mice fed a high fat diet over four weeks found that olanzapine administration exacerbated hyperglycemia the most in the mice that were obese at baseline [30]. This suggests that obesity prior to beginning SGA therapy may compound the metabolic effects of the SGA in a direct way, although human studies replicating this finding are needed.

Genetic polymorphisms in the methylation pathway are yet another reason why some individuals are more likely to develop MetS [22]. One such polymorphism is methylenetetrahydrofolate reductase (MTHFR) with those who break down folate more slowly being at higher risk of obesity [22].

SGAs are also known to affect the gut microbiota, which is thought to lead to gut dysbiosis and weight gain [31,32,33,34]. Animal models demonstrate that fecal transplantation from risperidone-treated mice into wild-type mice causes a reduction in basal metabolic rate and significant weight gain [30]. In another study of olanzapine-treated mice, the administration of a high-dose probiotic reversed weight gain and insulin resistance [32]. In yet another study, a prebiotic mixture was shown in rats to attenuate olanzapine-induced weight gain and increase the Bacteroidetes composition in the gut, while decreasing the Firmicutes composition [33]. Bahr et al. showed an altered Firmicutes/Bacteroidetes ratio which seems to promote obesity [34]. When this analysis was repeated with male youth treated with long-term risperidone, the boys who gained the most weight had significantly fewer Bacteroidetes compared to controls [31]. It is thought that the balance between the short chain fatty acids (SCFAs) produced by these two bacterial species plays a role in fat deposition in adipose tissue [34].

Another factor that may be implicated in SGA-induced weight gain is the body’s intracellular storage of iron, as measured by ferritin [35,36,37]. Low ferritin, indicating low bodily iron stores, is possible independent of anemia as measured by low hemoglobin and hematocrit [37]. This could be particularly relevant in youth whose bodies are growing and require sufficient intracellular iron stores [37]. Combining vertical growth with weight gain can deplete bodily iron stores more rapidly [37]. Studies by Calarge et al. demonstrate in children an inverse relationship between ferritin level decreases early in SGA therapy and more significant weight gain [35,36,37]. These results were independent of dietary iron intake, which fell within the recommended daily allowance for the study participants.

## 4. Managing Metabolic Syndrome in the Setting of SGA Prescription

There are several evidence-based, non-pharmacologic and non-surgical recommendations to prevent and manage MetS. These recommendations are meant to be suggestions for the clinician.

At the beginning of treatment:Prevention. Since obesity is difficult to treat in all populations, the axiom holds true that “an ounce of prevention is worth a pound of cure.” To that end, efforts have been made to reduce antipsychotic prescribing in youth and to clarify acceptable use of these medications, stressing trialing alternatives first, and to the development of guidelines around metabolic monitoring while on antipsychotics [7]. Some of these efforts have been successful, especially amongst patients of lower socioeconomic status—who perhaps are at the highest risk of obesity and MetS [38,39]. However, there are still situations in which the potential benefits of antipsychotic prescribing outweigh the potential risks, thus justifying ongoing SGA prescription. When possible, olanzapine and clozapine should be avoided due to these two agents having the greatest propensity to cause significant weight gain [22].Provide thorough informed consent to the parent/guardian and the patient about the SGA and the extent of the metabolic risks [17]. While there are not formal guidelines about risk stratification and SGA prescription in youth, the following suggestions can be extrapolated from existing studies about risk factors for pediatric obesity in general. As part of the consent process, obtain a thorough history, including personal and family history of MetS, hypertension, dyslipidemia, obesity, diabetes, and screen for food insecurity to help identify those who may be at higher risk of developing MetS. It is possible to help address some of the risk factors preventatively.Consider engaging the patient and family to make a weight management plan [26]. Utilize the tenets of motivational interviewing to discuss health goals and the plan to get there, including physical activity, food choices, screen time, and sleep.For patients with specific risk factors, consider addressing those risk factors in addition to a general weight management plan [39]. Screening for food insecurity and referral to appropriate community resources may be necessary, as food insecurity is correlated with higher risk of childhood obesity [40,41]. Children with high adverse childhood experience (ACE) scores or significant family dysfunction have higher rates of obesity [40]. Referral to family therapy or facilitating engagement in additional services may be helpful [26]. Families with higher intake of processed foods or children with extremely restricted diets may benefit from a referral to a nutritionist. Children who are physically inactive should be encouraged to be more active.Obtain the recommended baseline metabolic parameters, including baseline BMI, blood pressure, fasting glucose, and lipid panel. These are recommended by the American Diabetes Association (ADA), and the American Association of Child and Adolescent Psychiatry (AACAP) agrees with these recommendations [7,42]. Per American Psychiatric Society—American Diabetic Society consensus guidelines, weight should be checked every four weeks for the first three months of therapy; blood pressure and fasting lipids and blood glucose should again be checked after three months of therapy; fasting blood glucose should be obtained annually thereafter and fasting lipids every five years. Many clinicians check fasting blood glucose (or hemoglobin A1c %) and lipids yearly.“Start low and go slow” with dosing of the SGA [7].

Early in treatment:Monitor early weight gain carefully [26]. There are some data in adults suggesting that weight gain early in therapy is predictive of continued weight gain throughout treatment [43]. If significant early weight gain occurs, defined in the cited study as >1 kg in the first two weeks of therapy, forming a comprehensive weight management plan with the patient and family is especially important.Continue to evaluate risks and benefits of discontinuing the SGA where possible to limit the duration of use [7,26].Monitor and attempt to reduce co-prescription of mood stabilizers, antidepressants, antihistamines, and other medications known to cause weight gain [7].For patients requiring ongoing SGA therapy, continue to periodically review contributing lifestyle factors discussed in this paper and work on specific goals. Consider using the mnemonic “SMART” (specific, measurable, achievable, relevant, time-limited) to help the patient set and achieve their goals.Consider targeted, evidence-based supplementation for various aspects of MetS that may develop as a result of SGA therapy, the evidence for which is discussed below.

### 4.1. Recommendations for Making Better Food Choices

#### 4.1.1. Mediterranean Diet

The American Heart Association (AHA) recommends a Mediterranean-type diet to youth in general and specifically for those who are obese [44]. This dietary pattern emphasizes the intake of vegetables, fruits, whole grains, low-fat and nonfat dairy products, means, fish, and lean meat [44]. Youth who adhere the closest to a Mediterranean diet have lower rates of obesity and MetS [45,46]. The Mediterranean diet can be a helpful intervention in the treatment of non-alcoholic fatty liver disease (NAFLD) [47]. For younger children, using a traffic light dietary pattern may be helpful in dividing foods into “green light” (fruits, vegetables), “yellow light” (grains), and “red light” (least nutrient-dense, highest calories), and limiting “red light” foods to a certain number per week may be helpful [48].

As an additional benefit, the Mediterranean diet is likely protective against depression [49,50]. In adults with MetS, dietary patterns closest to the Mediterranean diet are correlated with lower rates of depressive symptoms [49]. Diets high in processed food and low in vegetables and fruits—the opposite of the Mediterranean diet—are predictive of depressive symptoms in youth as well [50].

#### 4.1.2. Dietary Fiber Consumption

Increased fiber consumption is linked to improved insulin sensitivity and to lower hemoglobin A1c percentages in patients with diabetes and pre-diabetes [51]. An increase in dietary fiber is one of the first-line interventions in the treatment of NAFLD as well [47]. For children, to calculate the recommended daily fiber intake, clinicians can use the “age plus 5” formula [48]. Fiber intake for a ten-year-old child, for example, should be 15 g daily [48]. By the time children reach age 15, their fiber intake should be at adult levels [48]. Assuming an average fiber intake of around 20 g daily for an adult, Reynolds et al. recommended from their findings that increasing daily fiber intake to 35–40 g daily significantly improved insulin sensitivity [51]. A good way to get more fiber is to simply replace refined grain products in the diet with whole grain products [51].

#### 4.1.3. Low-Glycemic Foods

Consumption of low-glycemic foods reduces post-prandial glucose rise and insulin spikes, resulting in better diabetes control according to a recent meta-analysis [52]. Specific suggestions might include substituting oatmeal for cereal at breakfast, brown rice for white rice, sweet potatoes for white potatoes, incorporating at least one vegetable at each meal, and eliminating processed foods like chips, candy, etc. [48]. Teaching patients to read food labels to identify the addition of sugars into sauces and dressings and to avoid products containing high fructose corn syrup (HFCS) as regular consumption of HFCS is linked with obesity and MetS [53,54,55]. 

#### 4.1.4. Reduce Intake of Sugar-Sweetened Beverages

Sugar-sweetened beverages (SSBs) (e.g., sodas and juices) are clearly associated with obesity in children and adolescents [16]. A Cochrane Review examined the efficacy of various interventions in reducing the consumption of SSB [56]. Interventions in that review with the highest evidence included food labeling, price increases on SSBs, restricting SSBs from the food stamp program—all of which indicate the importance of food policy in combating pediatric obesity. On an individual level, one of the best interventions from the Cochrane review to help obese adolescents reduce SSB consumption is for parents to put low-calorie drink options in their home to replace SSBs.

#### 4.1.5. Breakfast and Food Timing

Some evidence points to a correlation between obesity risk and breakfast skipping in children and adolescents [48]. Eating breakfast may lead to less snacking and smaller portion sizes at meals during the rest of the day [48]. Research is emerging about the relationship between obesity and the timing of meals in general, the regular scheduling of mealtimes, ranging from time of day a meal is consumed to time-restricted fasting; however, there are not pediatric-focused studies on these topics to our knowledge [57,58,59,60]. The macronutrient profile of food eaten prior to bedtime may also play a role in metabolism and the development of obesity, according to at least one mouse study involving concentrating carbohydrate intake before sleep. In this study, obese mice fed most of their daily carbohydrates prior to bedtime ate less, were leaner, and lost fat mass compared to obese mice fed carbohydrates consistently throughout the day [61]. Designing a follow-up study in human subjects, and then in youth, is needed.

#### 4.1.6. Mindfulness at Meals

Mindful eating can simultaneously increase food-related pleasure and reduce total calories consumed and reduce emotional eating [62]. However, studies are needed examining the effects of mealtime mindfulness interventions in obese youth.

### 4.2. Recommendations for Increasing Physical Activity

Increasing physical activity is near the top of the list of interventions in the treatment of pediatric obesity, second only to dietary intervention [48]. Many studies have shown that increasing physical activity for youth leads to a reduction in BMI [48]. Combining aerobic training and resistance training over a longer time period leads to the most significant reductions in BMI in youth according to a recent meta-analysis [63]. The CDC recommends that children and adolescents engage in 60 min of moderate-to-vigorous physical activity daily. However, that is not the reality for the majority in the United States, regardless of obesity status; only an estimated 24% of youth achieve that level of physical activity [64].

Instead of using the word “exercise”, we suggest using words like “movement” to further generalize the concept of being more active, which can include dancing, yoga, walking, stretching. One of the first steps in becoming more physically active might be reducing sedentary time, which tends to mean screen time for children and adolescents: in 2010, children and adolescents spent more than seven and a half hours daily in front of a screen [65].

As an additional benefit, physical activity is associated with fewer depressive symptoms and functions as a treatment for adolescent depressive symptoms of moderate effect size according to several meta analyses [66,67].

### 4.3. Recommendations for Healthy Sleep

Sleep disturbances are very common amongst children and adolescents with psychiatric disorders and are often at the core of the diagnosis [68]. Sleep disturbances also have metabolic consequences. The link between insufficient sleep and obesity and type 2 diabetes is well-established, both in adult and pediatric literature [69,70]. Short sleep duration conferred a 58% higher risk for obesity or overweight in one meta-analysis [71]. Authors also found a 92% increase in risk of obesity or overweight amongst children with the shortest sleep duration compared to children who slept longer. Most strikingly, for each additional hour of sleep, there was a 9% reduction in the risk of overweight or obesity [71]. Another study found that, after controlling for demographic factors, later adolescent bedtimes were associated with higher total daily fat intake, later first and last meals of the day, and greater after-dinner intake [72]. Conversely, earlier bedtimes are particularly helpful in reducing caloric intake in children and adolescents [73,74]. Guidelines from the National Sleep Foundation, endorsed by the American Academy of Pediatrics, suggest 10–13 hours (h) of sleep for preschoolers (ages 3–5), 9–11 h for school age children (6–13), and 8–10 h for teenagers (14–17). In addition to sufficient sleep and earlier bedtimes, additional sleep hygiene improvements may be indicated such as creating a soothing bedtime routine, restricting access to electronic devices within two hours of bedtime, limiting naps, keeping the sleeping environment dark and cool, and ensuring the youth uses their bed for sleep only [54,75].

Clinicians should also be especially attuned to medical conditions interfering with sleep as sleep disorders are also more common in patients with psychiatric disorders. Obstructive sleep apnea in the pediatric population is associated with higher risk of diabetes independent of obesity [76]. Restless leg syndrome is more prevalent in obese youth and can negatively impact sleep quality as well [77].

### 4.4. Recommendations for Stress Management

While mindfulness interventions related to prevention of obesity are in their infancy, one pilot trial for at-risk adolescents in a mindfulness group versus a health education control group did not find any evidence that BMI measurements were different between the two groups at the end of the six-week intervention [78]. Other mindfulness interventions such as mindfulness-based eating awareness training for the individual and/or family, breathwork training, and school-based mindfulness-based stress reduction (MBSR) training, targeting eating behaviors in adolescents have showed promising results [79].

### 4.5. Evidence-Based Supplementation for Metabolic Syndrome

#### 4.5.1. Vitamin D

Hypovitaminosis D is a predictor of cardiovascular mortality. In children, optimizing vitamin D is correlated with improvements in lipid profile, specifically in improving HDL [80,81]. Endocrine Society Guidelines recommend 25(OH)D levels ≥30 ng/mL [82].

#### 4.5.2. Omega 3 Fatty Acids for Hypertriglyceridemia

Supplementing with omega 3 fatty acids in the form of fish oil (3 grams daily) was found in a recent randomized, double-blind, placebo-controlled trial in children and adolescents to reduce serum triglycerides by 39% compared to 16% in placebo (*p* < 0.01) [83]. There are numerous other studies in the pediatric literature demonstrating similar results. In addition to triglyceride lowering properties, omega-3 supplementation has an evidence base in treating symptoms of ADHD, bipolar disorder, and depression, all of which can be off-label indications for SGA prescription [84,85,86,87,88].

#### 4.5.3. Alpha-Lipoic Acid (ALA) for Weight Loss

A recent randomized, double-blind, placebo-controlled trial of obese youth (*n* = 80) found that supplementation of ALA 300 milligrams twice daily for three months significantly improved BMI over the control group (*p* < 0.05) although lipid profiles did not improve [89]. It is known that ALA exerts anti-inflammatory benefits [89].

#### 4.5.4. Other Supplements

Probiotic supplementation, while shown to be helpful in attenuating weight gain in animal models as described above, does not yet have an evidence base in youth prescribed SGAs. Moreover, as described above, lower ferritin levels were correlated with risperidone-induced weight gain in boys; however, no studies show that supplementing with iron attenuates weight gain. Obese children tend to have lower serum magnesium levels, which was correlated in one study with lower insulin sensitivity [90]. No studies have examined magnesium supplementation in youth as a treatment for obesity or insulin resistance. Chromium supplementation in adults can reduce hemoglobin A1c%; however, there is not an evidence base for its use in the pediatric population [91]. Similarly, zinc supplementation may play a protective role against diabetes in adults, but studies are lacking in youth [92].

Myo-inositol has an evidence base in reducing BMI in women with polycystic ovarian syndrome (PCOS) and in obese patients with fasting plasma insulin levels but has not been studied in youth [93]. Herbs such as berberine and ginseng, which have an evidence base in adults with insulin resistance, have not been studied in the pediatric population and are thus not currently recommended [94,95].

## 5. Conclusions

The metabolic side effects of SGAs in youth are significant and can have far-reaching consequences into adulthood. Clinicians who prescribe SGAs should carefully weigh risks and benefits of prescribing, reserving prescription for cases in which the benefits clearly outweigh risks. We also recommend that clinicians educate patients and their families about metabolic risks of SGAs and that this education be ongoing, not just at the start of therapy. We suggest that clinicians take a motivational interviewing approach to identify modifiable lifestyle factors that are risks for the development of obesity and MetS and to create specific goals, taking into account various risk factors including medical history, family history, family lifestyle, and socioeconomic factors. Interventions with the most evidence include increasing consumption of fruits, vegetables, and whole grains, increasing physical activity, improving sleep, and fish oil supplementation. While outside the scope of this review, certain pharmacologic interventions such as metformin may be indicated as well; however, they are best done as adjunctive to lifestyle changes. Finally, the mental health benefits of treating obesity should not be overlooked. Treating obesity is associated with a reduction in depressive and anxious symptoms in the pediatric population [96].

Future directions for research should include a clearer definition of the exact cut-off values to establish the diagnosis of MetS; additional innovative and cost-effective behavioral intervention trials focusing on lifestyle choices; and the establishment of a more in-depth understanding of the interplay between chronic stress, trauma, and metabolism.

In conclusion, the prescription of an SGA offers the opportunity to work collaboratively with the patient and their family to improve overall health and to mitigate potential serious side effects—an opportunity that should not be missed.

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
