# Peer review of "Integrative Management of Metabolic Syndrome in Youth Prescribed Second-Generation Antipsychotics"

_medsci, 2020, doi:10.3390/medsci8030034_

Round 1

Reviewer 1 Report

GENERAL COMMENT

This submission addresses a topic of major clinical importance. Accuracy in writing, however should be improved. In addition, comments on significance and limitations of the notion of MetS should further be expanded. Principles of chrono-nutrition should briefly be alluded to. Finally, this submision would benefit from a better developed paragraph on research agenda as well as a graphical abstract.

SPECIFIC COMMENT

MAJOR

1. These Authors may preliminarily be willing to discuss whether physical inactivity and unhealthy diets occur as a manifestation of depression per se rather than as a side effect of drug treatment.

2. The following section is repetitive and destructured."Pediatric obesity is associated with numerous medical complications including type 2 diabetes, hypertension, dyslipidemia, nonalcoholic fatty liver disease, as well as the risk of cancer in adulthood [2]. Metabolic syndrome (MetS) is a term describing the relationship between central obesity, insulin resistance, hypertension, and dyslipidemia. Pediatric MetS is also strongly predictive of adult MetS and type II diabetes [3]." In reworking it, please highlight importance and limitations of the MetS concept (Lonardo A, Ballestri S, Marchesini G, Angulo P, Loria P. Nonalcoholic fatty liver disease: a precursor of the metabolic syndrome. Dig Liver Dis. 2015;47(3):181‐190).

3. Further to the importance of breakfast, please discuss also on the importance of food timing (Sofer S, Eliraz A, Madar Z, Froy O. Concentrating carbohydrates before sleep improves feeding regulation and metabolic and inflammatory parameters in mice. Mol Cell Endocrinol. 2015;414:29‐41. Ha K, Song Y. Associations of Meal Timing and Frequency with Obesity and Metabolic Syndrome among Korean Adults. Nutrients. 2019;11(10):2437. Published 2019 Oct 13. doi:10.3390/nu11102437. Singh RB, Cornelissen G, Mojto V, et al. Effects of circadian restricted feeding on parameters of metabolic syndrome among healthy subjects. Chronobiol Int. 2020;37(3):395‐402. Pot GK, Almoosawi S, Stephen AM. Meal irregularity and cardiometabolic consequences: results from observational and intervention studies. Proc Nutr Soc. 2016;75(4):475‐486. doi:10.1017/S0029665116000239Kosmadopoulos A, Kervezee L, Boudreau P, et al. Effects of Shift Work on the Eating Behavior of Police Officers on Patrol. Nutrients. 2020;12(4):999. Published 2020 Apr 4. doi:10.3390/nu12040999. Al Abdi T, Andreou E, Papageorgiou A, Heraclides A, Philippou E. Personality, Chrono-nutrition and Cardiometabolic Health: A Narrative Review of the Evidence [published online ahead of print, 2020 May 14]. Adv Nutr. 2020;nmaa051).

4. Throughout the manuscript:

- make sure to identfy all initialisms when used for the first time. For example CDC; FDA; ADHD; TNF-α, IL-6; IL-1β; BMI; 5-HT2C and so on

- rework Tipe II diabetes ---> Type 2 Diabetes

- abbreviations must be explained in the folowing order: explain the full wording first; next give the abbreviation. For example: "ACE scores (adverse childhood experiences)" should read "adverse childhood experiences (ACE) scores". Please, check the text accurately.

- Metabolic Syndrome has been shortened as MetS at line 26. Therefore, throught the manuscript these Authors may be willing to invariably use MetS rather than Metabolic Syndrome.

- Conversely, " nonalcoholic fatty liver disease" is first used in the introduction but is shortened as NAFLD only at line 180.

- Support relevant statemenst with appropriate references. For example "Low ferritin, indicating low bodily iron stores, is possible independent of anemia as measured by low hemoglobin and hematocrit." "Make a weight management plan with the patient and family. Utilize the tenets of motivational interviewing to discuss health goals and the plan to get there, including physical activity, food choices, screen time, and sleep."

5. Some authors have proposed the use of anti-depressants in NAFLD. Could these Authors comment on this proposal ?

6. I think research gaps should better be outlined. In particular, I would suggest developing a section on the limitations of the approach adopted here and a short discussion of the areas most in need to be pursued. In addition, a graphcal abstract would increase the potential for this study to be cited.

MINOR

1. the United States has --> have

Author Response

In response to "General Comment"-

Thank you for this valuable suggestion. We added the limitations of MetS as a concept and discussed briefly in the introduction paragraph. We also added brief content about chrono nutrition in the “breakfast” section.  In addition, comments on research limitations and future directions for research were added throughout the paper.

In response to Major Comments- 

1. This was discussed in the opening paragraph of the pathophysiology on MetS section. 

2. We appreciate the reference. We have added the reference and reworked the section. 

3. Thank you for the suggestion. Many of these references were discussed and incorporated.

4. Thank you. Addressed.  Type 2 diabetes changed.  Regarding referencing throughout the paper, thank you for pointing this out. We have reviewed and added references where possible.

5. The intent of this article is to provide overview of management using integrative medicine principles. Given the controversy about whether the NAFLD is a direct result of medications including antidepressants on the liver or a consequence of weight gain triggered by medication, we felt it was beyond the scope of our review to comment on the issue.

6. Thank you. We have highlighted areas for further research throughout the paper.

In response to Minor Comments-

1. The “United States” being the name of a singular country should keep the “to have” verb in the singular form unless we are mistaken.

Reviewer 2 Report

The manuscript reviews the metabolic side effects of second generation antipsychotics in children and adolescents. The authors then propose ongoing ways to manage metabolic effects in youth including preventative measures. This is a well written review on an important topic. I only have two minor suggestions that could be easily addressed. 

  1. Page 3 line 99: I believe the authors meant to write basal metabolic rate instead of basal metabolic weight. 
  2. line 105: would change "human boys" to male youth 

Author Response

Regarding the first point, thank you, we addressed this.

Regarding the second point, thank you again, we also changed this.

Reviewer 3 Report

Rice and Ramtekkar have written a concise and timely review article highlighting the utility of non-pharmaceutical lifestyle interventions in treating and preventing SGA-induced metabolic syndrome in youth. While there have been many papers that have highlighted pharmaceutical approaches in treating SGA-induced metabolic dysfunction, this review is novel as it focuses on the role of diet and physical activity/exercise as adjunct approaches. While SGA’s clearly cause weight gain and dysglycemia with chronic use, there is also a noted direct effect of SGAs on glucose metabolism, independent of weight gain. This has been shown in both rodent (1) and human studies (2) and should be discussed. Along a similar line the authors should briefly discuss the findings that chronic exercise training or an acute bout of exercise can protect against SGA-induced metabolic dysfunction (3, 4).

  1. Townsend LK, Peppler WT, Bush ND, & Wright DC (2017) Obesity exacerbates the acute metabolic side effects of olanzapine. Psychoneuroendocrinology 88:121-128.
  2. Hahn MK, et al. (2013) Acute effects of single-dose olanzapine on metabolic, endocrine, and inflammatory markers in healthy controls. Journal of clinical psychopharmacology 33(6):740-746.
  3. Boyda HN, et al. (2014) Routine exercise ameliorates the metabolic side-effects of treatment with the atypical antipsychotic drug olanzapine in rats. Int J Neuropsychopharmacol 17(1):77-90.
  4. Castellani LN, Peppler WT, Miotto PM, Bush N, & Wright DC (2018) Exercise Protects Against Olanzapine-Induced Hyperglycemia in Male C57BL/6J Mice. Sci Rep 8(1):772.

Author Response

Thank you for these suggestions. These four references were incorporated in a section expanded to look at SGAs and hyperglycemia.

Round 2

Reviewer 1 Report

Most of the points raised have been addressed.